# Comparative Analysis of Metaheuristic and Heuristic Strategies in Forest Fire Suppression

## Abstract

Forest fires represent a significant and escalating global threat, necessitating the development of effective suppression strategies. This paper investigates the application of computational intelligence, specifically comparing a metaheuristic approach, Ant Colony Optimization (ACO), with a simpler heuristic, a Greedy algorithm, for the strategic placement of firebreaks. Although metaheuristics like ACO are generally anticipated to yield superior solutions for complex optimization problems, simulation results under a specific, constrained scenario—a centrally located fire on a 20x20 grid with a high density of firebreaks—demonstrate that the Greedy strategy unexpectedly outperformed ACO in both minimizing the area burned and the time required for containment. This report analyzes this counterintuitive outcome, providing theoretical explanations grounded in the principles of local versus global optimization and contextualizing the findings within the broader optimization literature.

## 1 Introduction

Forest fires, exacerbated by global climate change, pose an increasing threat to ecological systems, human populations, and economic stability globally. The escalating frequency and intensity of these catastrophic events necessitate the development of highly effective and efficient suppression strategies. Traditional firefighting methods, often reliant on expert judgment and reactive decision-making, frequently contend with cognitive biases, incomplete information, and the inherent complexity of dynamic fire behavior, leading to suboptimal resource allocation and increased operational risks. Computational intelligence, particularly metaheuristic algorithms, offers a promising avenue for overcoming these limitations by providing robust solutions to complex, dynamic optimization problems through extensive exploration of vast solution spaces (Carta et al., 2023). Among these, Ant Colony Optimization (ACO), inspired by the collective foraging behavior of ants, has demonstrated significant potential for identifying optimal paths in dynamic networks. This paper previously introduced a conceptual framework that leverages ACO to strategically deploy firefighting resources, with the aim of minimizing overall fire damage by optimizing firebreak placements. For comparative evaluation, a simpler Greedy algorithm was also implemented. Although metaheuristics like ACO are generally expected to outperform simpler heuristics in complex problems due to their global optimization capabilities, recent simulations reveal a scenario where the Greedy strategy achieved superior performance in terms of both area burned and containment time. This report provides a detailed analysis of this outcome, offering theoretical explanations and contextualizing the findings within the broader literature on local versus global optimization in spatial containment problems.

Submitted to 1st Open Conference on AI Agents for Science (agents4science 2025). Do not distribute.

## 2 Background on optimization algorithms

### 2.1 Metaheuristics: ant colony optimization (ACO)

Metaheuristics are high-level frameworks designed to guide the search for solutions to optimization problems, allowing them to escape local optima and explore larger solution spaces. These algorithms are particularly effective for solving NP-hard problems, which are characterized by their computational intractability for exact solutions within practical timeframes. Ant Colony Optimization (ACO), a prominent metaheuristic, derives its inspiration from the ability of ant colonies to find the shortest paths between their nest and food sources through pheromone deposition. In the context of fire suppression, virtual "ants" construct paths representing potential firebreak locations. The probability of an ant at node $i$ choosing to move to node $j$ is determined by the pheromone level ($\tau_{ij}$) and heuristic information ($\eta_{ij}$), governed by Equation 1:

$$P_{ij} = \frac{(\tau_{ij}^{\alpha})(\eta_{ij}^{\beta})}{\sum_{k \in \text{allowed}}(\tau_{ik}^{\alpha})(\eta_{ik}^{\beta})} \tag{1}$$

where $\alpha$ and $\beta$ are parameters that control the influence of the pheromone trail and heuristic information, respectively. The heuristic information, $\eta_{ij}$, is a composite value based on factors like fuel load and proximity to the fire. Pheromone levels are updated iteratively based on the quality of the solutions found, encouraging convergence towards optimal paths. This adaptive mechanism enables ACO to conduct global searches, making it a robust choice for complex, dynamic problems.

### 2.2 Simple heuristics: the Greedy approach

In contrast to metaheuristics, simple heuristics, such as the Greedy algorithm, make decisions based on immediate, local information at each step without considering the global implications of these choices. A Greedy algorithm selects the locally optimal choice at each stage with the expectation of finding a global optimum. For instance, in firebreak placement, our Greedy strategy prioritizes nodes based on their Euclidean distance to the fire's origin. The distance $d$ between a node $(x_1, y_1)$ and the fire start $(x_f, y_f)$ is calculated as shown in Equation 2:

$$d = \sqrt{(x_1 - x_f)^2 + (y_1 - y_f)^2} \tag{2}$$

While computationally less intensive and faster, Greedy algorithms are susceptible to converging to local optima. Despite this, they can be highly effective in specific scenarios, particularly when the problem structure favors local decisions.

## 3 Simulation methodology and scenario design

### 3.1 Forest model and fire spread dynamics

The forest environment is modeled as a discrete, grid-based graph, $G = (V, E)$, on a $20 \times 20$ grid (Alexandridis et al., 2011). The fire spread model is probabilistic. A burning node can ignite an unburned adjacent neighbor with an ignition probability, $P_{\text{ignite}}$, determined by the neighbor's fuel load ($f_l$) and a wind factor ($w_f$), as defined in Equation 3:

$$P_{\text{ignite}} = \frac{f_l + w_f}{2.0} \tag{3}$$

The `wind_factor` was set to 0.8. Nodes designated with a firebreak status cannot burn, representing defensive lines. The simulation terminates after 50 time steps or when the fire is contained.

### 3.2 Firebreak placement strategies implemented

- **Ant colony optimization (ACO):** The ACO algorithm seeks an optimal set of 25 firebreak nodes. The objective is to minimize a weighted cost function combining burned area ($A_{burned}$) and containment time ($T_{contain}$), shown in Equation 4:

$$\text{Cost} = w_1 \cdot A_{burned} + w_2 \cdot T_{contain} \tag{4}$$

where $w_1 = 0.7$ and $w_2 = 0.3$. The heuristic information incorporates the inverse of a node's fuel load and its proximity to the fire start. A beta parameter of 1.5 was used to emphasize this heuristic information.

- **Greedy algorithm:** The Greedy algorithm selects 25 firebreak nodes based on their Euclidean distance to the fire's starting point, prioritizing the closest nodes to encircle the fire rapidly.

### 3.3 Specific scenario parameters and implications - scenario 1

The simulation was configured with a unique, constrained scenario:

- **Grid size:** A 20x20 grid (400 nodes).
- **Fire start position:** The fire initiates at the center of the grid (10, 10), creating a symmetric problem.
- **Number of firebreaks:** 25 firebreaks are allocated (6.25% of total nodes), a high density for containment.
- **Wind factor:** A moderate `wind_factor` of 0.8.
- **Fuel load distribution:** Randomly assigned fuel loads between 0.1 and 1.0.

### 3.4 Specific scenario parameters and implications - scenario 2

After conducting the first scenario, the need for a second experiment was identified(see section 4). Thus, an altered version of the first experiment was reexamined.

- **Grid size:** A 20x20 grid (400 nodes).
- **Fire start position: Fire starts at random position in the 20x20 grid.**
- **Number of firebreaks:** 25 firebreaks are allocated (6.25% of total nodes), a high density for containment.
- **Wind factor:** A moderate `wind_factor` of 0.8.
- **Fuel load distribution:** Randomly assigned fuel loads between 0.1 and 1.0.

### 3.5 Specific scenario parameters and implications - scenario 3

After conducting the second scenario, the need for a third experiment was identified(see section 4). Thus, an altered version of the second experiment was reexamined.

- **Grid size:** A 20x20 grid (400 nodes).
- **Fire start position: 7 different fires each start at random positions in the 20x20 grid.**
- **Number of firebreaks:** 25 firebreaks are allocated (6.25% of total nodes), a high density for containment.
- **Wind factor:** A moderate `wind_factor` of 0.8.
- **Fuel load distribution:** Randomly assigned fuel loads between 0.1 and 1.0.

### 3.6 Control of stochastic error

Since the ACO function creates stochastic error by the random functions, the experiment was conducted 100 times, and mean data was extracted from experiments.

## 4 Simulation results and analysis

### 4.1 Simulation results - scenario 1

Execution of the simulation under the specified scenario reveals a notable difference in performance between the ACO and Greedy strategies.

Table 1: Comparison of fire suppression strategies(100 executions)

| Metric | ACO Strategy | Greedy Strategy |
|---|---|---|
| Total Area Burned (Average, nodes) | 88.74 | 1.00 |
| Total Area Burned (Standard Deviation, nodes) | 147.90 | 0.00 |
| Time to Containment (Average, steps) | 10.97 | 1.00 |
| Time to Containment (Standard Deviation, steps) | 16.95 | 0.00 |

As detailed in Table 1, the Greedy strategy significantly outperformed the ACO strategy. The Greedy algorithm limited the total area burned to a single node and achieved containment within one time step. In contrast, the ACO strategy resulted in approximately 89 nodes burned and required 11 steps for containment, with a high margin of error. As later mentioned in section 5, the highly symmetric design of the map was suspected as the cause of error.

## 4.2 Simulation results - scenario 2

Execution of the simulation under the specified scenario reveals no difference in the heuristic Greedy algorithm, while the data for ACO showed some improvement.

Table 2: Comparison of fire suppression strategies(100 executions)

| Metric | ACO Strategy | Greedy Strategy |
|---|---|---|
| Total Area Burned (Average, nodes) | 45.60 | 1.00 |
| Total Area Burned (Standard Deviation, nodes) | 114.31 | 0.00 |
| Time to Containment (Average, steps) | 6.09 | 1.00 |
| Time to Containment (Standard Deviation, steps) | 12.70 | 0.00 |

As detailed in Table 2, it is clearly visible that, although the ACO method did improve in efficiency, it is still far behind the efficiency of the simple Greedy method, which hasn't changed in value. Thus, it was clear that the heuristic algorithm was too optimized for a single-fire task. Since the Greedy algorithm can simply "surround" the fire source with 4 firebreaks, any one-fire case can be easily handled by the Greedy algorithm.

## 4.3 Simulation results - scenario 3

Execution of the simulation under the specified scenario reveals drastically different data, but no difference in trend; the Greedy algorithm outperforms the ACO algorithm.

Table 3: Comparison of fire suppression strategies(100 executions)

| Metric | ACO Strategy | Greedy Strategy |
|---|---|---|
| Total Area Burned (Average, nodes) | 340.50 | 147.46 |
| Total Area Burned (Standard Deviation, nodes) | 31.16 | 164.42 |
| Time to Containment (Average, steps) | 24.63 | 17.21 |
| Time to Containment (Standard Deviation, steps) | 6.49 | 18.83 |

As detailed in Table 3, since the number of fire sources exceeds 6($\lfloor \frac{25}{4} \rfloor$), we can know that fire sources must be connected by a side in order to be contained. This has caused a drastic change in data, leading to large error in the Greedy strategy. The ACO strategy can be seen taking care of the situation without much deviation from the mean, however takes too long to contain and loses a lot of land.

The Greedy strategy has preformed much worse compared to scenarios 1 and 2 in secnario 3. This is most likely due to the randomness of fire origins requiring a more global resource distribution.

# 5 Explaining Greedy's superior performance

The observed performance of the Greedy algorithm can be attributed to the fundamental dichotomy between local and global optimization strategies, coupled with the unique characteristics of the problem instance.

## 5.1 Local vs. global optimization principles

Optimization problems typically involve finding the global optimum among all feasible solutions. Greedy algorithms inherently pursue local optima, making the best choice at each step based on immediate information. This approach is computationally efficient but does not guarantee a globally optimal solution. Metaheuristics, including ACO, are designed to traverse the solution space more thoroughly, balancing exploration and exploitation to find global optima.

## 5.2 Specific factors contributing to Greedy's success

- **Centralized Fire Start and Symmetric Problem Structure:** The fire originating at the center of the grid creates a highly symmetric problem where the most effective containment strategy is to establish a perimeter in adjacent nodes. The Greedy algorithm's focus on proximity aligns perfectly with this optimal initial move.
- **High Density of Firebreaks:** With 25 firebreaks on a 400-node grid, a significant proportion of the area can be converted into containment lines. This abundance allows even a simple proximity-based strategy to quickly form an effective perimeter.
- **Greedy Heuristic's Direct Relevance:** The Greedy heuristic of selecting nodes by Euclidean distance was optimally aligned with the ideal strategy for this specific centralized fire scenario.
- **ACO's Exploration Overhead:** ACO inherently incurs computational overhead for exploration and pheromone updating. In a scenario where the optimal solution is immediate and localized, ACO's broader search may delay the concentration of resources on the most critical nodes.

## 5.3 Literature context for simpler heuristics outperforming metaheuristics

The observed outcome is not an anomaly but is well-documented in the optimization literature. Some research indicates that simpler local search heuristics can prove highly competitive or even superior to more complex metaheuristics, especially when constraints limit the effective search space. This phenomenon underscores the importance of aligning algorithm selection with the problem's underlying structure. The simulation results confirm this understanding, illustrating that for problems with inherent symmetry and localized optimal solutions, a Greedy strategy can indeed be more effective.

# 6 Limitations of the simulation and interpretation

While the simulation provides valuable insights, it is important to acknowledge its inherent limitations:

- **Model simplifications:** The 20x20 grid is a significant abstraction of a real-world forest.
- **Stochastic nature:** The stochastic property of the simulation generates large error. Robust conclusions will require more efficient algorithms.
- **Simplified objective function:** The ACO's objective function is a simplification of real-world fire suppression costs.
- **Scenario specificity:** The most crucial limitation is the high degree of scenario specificity. The results may not be generalizable to all forest fire scenarios where the optimal solution is not readily apparent.

# 7 Future work and broader implications

The findings underscore several important directions for future research:

- **Extensive scenario diversity:** Future experiments must include a broader range of scenarios (e.g., varying fire start locations, grid topologies) to test algorithms across a spectrum of challenges.
- **Comprehensive parameter sensitivity analysis:** A systematic analysis of ACO's parameters is essential for robust configurations.
- **Hybrid approaches:** Exploring hybrid algorithms that combine Greedy's speed for initial containment with ACO's strategic optimization for evolving fires is a promising avenue (Aranzazu-Suescun et al., 2014).
- **Integrating real-world complexity:** Future research should incorporate more sophisticated models for fire spread, accounting for dynamic wind, heterogeneous fuel types, and topography.

Despite its performance in this constrained scenario, ACO's potential for complex, large-scale forest fire problems remains highly relevant. Its ability to identify non-obvious, globally optimal paths offers a significant advantage over local methods in truly challenging environments.

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

# A    Appendix : code and experiment reproducibility

This appendix provides the necessary details to reproduce the experiments in the paper. Necessary code is included in the supplementary material. The supplementary repository can be accessed at : https://github.com/codingneerd/forestfire.

## A.1    System requirements and dependencies

The simulation was executed on a standard personal computer and does not require specialized hardware. It has, however, been tested on the following two devices :

- **Device:** Samsung Galaxy Book 5 Pro 360
- **CPU:** Intel(R) Core(TM) Ultra 7 256V
- **GPU:** Intel(R) Arc(TM) 140V GPU (8GB)
- **RAM:** 16GB
- **OS:** Windows 11 Home

- **Device:** Apple Macbook Air 15" (2025)
- **CPU:** Apple Silicon M4
- **GPU:** Apple Silicon M4
- **RAM:** 16GB
- **OS:** MacOS Sequoia

The script is written in Python v3.9.10 and relies on the following major libraries:

- **NumPy:** For numerical operations, particularly distance calculations.
- **NetworkX:** To create and manage the grid-based graph representing the forest.
- **Matplotlib:** For plotting the results.

To ensure compatibility, it is recommended to install the specific versions of these packages using the following command:

```
pip install numpy networkx matplotlib
```

## A.2    Execution instructions

To run the full simulation comparing the Ant Colony Optimization (ACO) and Greedy strategies, each scenario file is provided under the naming convention

```
scenario-n.py
```

and execute it from your terminal with the following command:

```
python scenario-n.py
```

The script will print the final comparison of the total burned area and the time to containment for both strategies to the console, along with the standard deviation of each value.

## A.3    Algorithm and simulation parameters

All parameters used in the code are separated as variables on the top of the code. Altering these values will result in different simulation results.


