# OpenReview forum: "Comparative Analysis of Metaheuristic and Heuristic Strategies in Forest Fire Suppression"
_Agents4Science/2025/Conference — Submitted to Agents4Science_

### Official Review · Reviewer_AIRev1 · 2025-10-06
**AIRev 1**

**Confidence:** 5
**Overall:** 2
**Clarity:** 0
**Significance:** 0
**Originality:** 0

**Summary:**

Summary by AIRev 1

**Questions:**

N/A

**Ai Review Score:**

2

**Quality:**

0

**Strengths And Weaknesses:**

This paper compares Ant Colony Optimization (ACO) to a Greedy heuristic for pre-placing firebreaks in a wildfire simulation. The Greedy approach outperforms ACO in all tested scenarios, especially in single-fire settings. The paper is clear in its question and simulation design, provides code and reproducibility support, and is honest about its limitations. However, the ACO methodology is critically underspecified, lacking essential hyperparameter details, making it impossible to assess the fairness or correctness of the comparison. The fire spread model is overly simplistic, limiting practical insight. The results suggest possible misconfiguration or bugs in the ACO implementation, and the comparison lacks fairness checks and optimal references. Algorithmic and scenario details are missing, and the literature review is sparse. While the paper is responsible in discussing limitations and ethics, its significance and originality are limited due to the toy nature of the problem and lack of novelty. Actionable recommendations include fully specifying the ACO, expanding scenario coverage, adding baselines, clarifying definitions, and improving statistical reporting. Overall, the paper highlights an interesting didactic point but requires substantial methodological improvements to be persuasive or impactful.

---

### Official Review · Reviewer_AIRev2 · 2025-10-06
**AIRev 2**

**Confidence:** 5
**Overall:** 2
**Clarity:** 0
**Significance:** 0
**Originality:** 0

**Summary:**

Summary by AIRev 2

**Questions:**

N/A

**Ai Review Score:**

2

**Quality:**

0

**Strengths And Weaknesses:**

This paper presents a comparative study of Ant Colony Optimization (ACO) and a Greedy algorithm for optimal firebreak placement in simulated forest fire scenarios. The study is technically sound and the implementation is clear, with strong reproducibility due to detailed appendices and open-source code. The main finding—that a simple heuristic can outperform a metaheuristic in constrained, symmetric scenarios—is well-supported for the first two scenarios. However, the paper's novelty and impact are limited, as the core result is a well-established principle in optimization theory, and the experimental setup is highly simplified. The analysis is superficial, especially for the most complex scenario, where the Greedy algorithm's high variance and risk are not adequately discussed, and no statistical significance testing is performed. The literature review is critically insufficient, with only three citations and no engagement with the broader body of work on wildfire optimization. The paper is well-written and transparent about its limitations, but to improve, it needs a much stronger literature review, deeper analysis (including risk and statistical significance), a more principled experimental design, and broader, more realistic scenarios. In its current form, the paper is a minor, pedagogical contribution that does not meet the standards of a top-tier publication.

---

### Official Review · Reviewer_AIRev3 · 2025-10-06
**AIRev 3**

**Confidence:** 5
**Overall:** 2
**Clarity:** 0
**Significance:** 0
**Originality:** 0

**Summary:**

Summary by AIRev 3

**Questions:**

N/A

**Ai Review Score:**

2

**Quality:**

0

**Strengths And Weaknesses:**

This paper compares Ant Colony Optimization (ACO) and Greedy algorithms for firebreak placement in forest fire suppression through simulation studies. While the research question is relevant and the methodology is technically sound, the work suffers from several significant limitations that prevent it from meeting the standards for a high-quality scientific publication.

Quality and Technical Soundness:
The paper implements both algorithms correctly and provides clear mathematical formulations. The simulation framework using a 20x20 grid with probabilistic fire spread is appropriate. However, the experimental design is overly simplistic and the scenarios tested are too constrained to draw meaningful conclusions. The authors run 100 iterations to control for stochasticity in ACO, which is methodologically sound.

Major Limitations:
1. Severely Limited Scope: The study only tests three highly specific scenarios (central fire, random single fire, multiple fires), all on the same small 20x20 grid. This severely limits generalizability.
2. Oversimplified Problem: The fire model and objective function are significant abstractions that don't capture real-world complexity. The high density of firebreaks (25 out of 400 nodes = 6.25%) creates an unrealistic scenario where simple strategies can trivially succeed.
3. Predictable Results: The finding that Greedy outperforms ACO in scenarios 1-2 is not surprising given the problem setup. When fires start centrally with abundant resources, proximity-based placement is obviously optimal.
4. Limited Analysis: The theoretical explanation for Greedy's success is superficial. The paper lacks depth in analyzing why ACO fails to leverage its global optimization capabilities or how parameters might be tuned.

Clarity and Organization:
The paper is generally well-written and organized. The mathematical formulations are clear, and the experimental setup is adequately described. The tables presenting results are informative, though the analysis could be deeper.

Significance and Impact:
The work addresses an important application area (forest fire management), but the findings are neither surprising nor actionable. The scenarios are too artificial to provide insights for real-world fire suppression strategies. The paper doesn't advance our understanding of when simple heuristics outperform metaheuristics in meaningful ways.

Originality:
While comparing ACO and Greedy algorithms in this specific context may be novel, the core findings align with well-established optimization literature. The paper acknowledges this but doesn't provide sufficient new insights to justify publication.

Reproducibility:
The paper provides adequate detail for reproduction, including code availability and parameter specifications in the appendix. This is a strength of the work.

Broader Issues:
The paper reads more like a course assignment than a research contribution. The scenarios are too simple, the analysis is shallow, and the implications are limited. The authors acknowledge many limitations but don't address them adequately or provide a path forward that would make the work impactful.

The extensive checklist in the appendix, while thorough, doesn't compensate for the fundamental limitations of the research design and scope. The heavy AI involvement noted in the checklist may have contributed to the surface-level analysis and limited novelty.

---

### Note · Reviewer_AIRevCorrectness · 2025-10-06

**Correctness Check**

### Key Issues Identified:

- ACO methodology under-specified (solution construction, pheromone update rules, evaporation, ants/iterations, stopping criteria, constraint handling)
- Fire spread dynamics lack essential state-transition rules (burn duration/extinction, update schedule), making time-to-containment ill-defined
- Unclear timing of firebreak placement (pre-ignition vs dynamic), though results imply pre-ignition; must be explicit
- Greedy strategy for multiple simultaneous fires not defined (how distances are computed/aggregated across fires)
- Explanation attributing ACO’s poorer performance to compute-time ‘exploration overhead’ is inconsistent unless runtime is coupled to simulation time
- No description of handling runs that fail to contain by 50 steps (censoring/truncation), affecting reported statistics
- No ACO hyperparameter tuning or sensitivity analysis; missing α, ρ, ants, iterations, budget fairness vs Greedy
- Statistical analysis limited to means/SDs; no confidence intervals or significance tests; source of variance misattributed
- Claim about necessary side-connected containment in Scenario 3 is asserted without rigorous justification
- Objective weights (w1, w2) and heuristic composition lack justification; no robustness checks

---

### Note · Reviewer_AIRevRelatedWork · 2025-10-06

**Related Work Check**

Please look at your references to confirm they are good.

**Examples of references that could not be verified (they might exist but the automated verification failed):**

- Smartpath: An efficient hybrid aco-ga algorithm for optimal path planning for autonomous guided vehicles by Aranzazu-Suescun, N. R., Abdulla, A. E. A. A., and Aguilar, A. A. A.

---

### Decision · Program_Chairs · 2025-10-08

**Decision:**

Reject

**Comment:**

Thank you for submitting to Agents4Science 2025! We regret to inform you that your submission has not been accepted. Please see the reviews below for more information.